# Sparsified SGD with Memory

**Sebastian U. Stich**      **Jean-Baptiste Cordonnier**      **Martin Jaggi**

Machine Learning and Optimization Laboratory (MLO)
EPFL, Switzerland

## Abstract

Huge scale machine learning problems are nowadays tackled by distributed optimization algorithms, i.e. algorithms that leverage the compute power of many devices for training. The communication overhead is a key bottleneck that hinders perfect scalability. Various recent works proposed to use quantization or sparsification techniques to reduce the amount of data that needs to be communicated, for instance by only sending the most significant entries of the stochastic gradient (top-$k$ sparsification). Whilst such schemes showed very promising performance in practice, they have eluded theoretical analysis so far.

In this work we analyze Stochastic Gradient Descent (SGD) with $k$-sparsification or compression (for instance top-$k$ or random-$k$) and show that this scheme converges at the same rate as vanilla SGD when equipped with error compensation (keeping track of accumulated errors in memory). That is, communication can be reduced by a factor of the dimension of the problem (sometimes even more) whilst still converging at the same rate. We present numerical experiments to illustrate the theoretical findings and the good scalability for distributed applications.

## 1 Introduction

Stochastic Gradient Descent (SGD) [29] and variants thereof (e.g. [10, 16]) are among the most popular optimization algorithms in machine- and deep-learning [5]. SGD consists of iterations of the form

$$\mathbf{x}_{t+1} := \mathbf{x}_t - \eta_t \mathbf{g}_t \,, \tag{1}$$

for iterates $\mathbf{x}_t, \mathbf{x}_{t+1} \in \mathbb{R}^d$, stepsize (or learning rate) $\eta_t > 0$, and stochastic gradient $\mathbf{g}_t$ with the property $\mathbb{E}[\mathbf{g}_t] = \nabla f(\mathbf{x}_t)$, for a loss function $f \colon \mathbb{R}^d \to \mathbb{R}$. SGD addresses the computational bottleneck of full gradient descent, as the stochastic gradients can in general be computed much more efficiently than a full gradient $\nabla f(\mathbf{x}_t)$. However, note that in general both $\mathbf{g}_t$ and $\nabla f(\mathbf{x}_t)$ are *dense* vectors[1] of size $d$, i.e. SGD does not address the communication bottleneck of gradient descent, which occurs as a roadblock both in distributed as well as parallel training. In the setting of distributed training, communicating the stochastic gradients to the other workers is a major limiting factor for many large scale (deep) learning applications, see e.g. [3, 21, 33, 44]. The same bottleneck can also appear for parallel training, e.g. in the increasingly common setting of a single multi-core machine or device, where locking and bandwidth of memory write operations for the common shared parameter $\mathbf{x}_t$ often forms the main bottleneck, see e.g. [14, 18, 25].

A remedy to address these issues seems to *enforce* applying smaller and more efficient updates $\text{comp}(\mathbf{g}_t)$ instead of $\mathbf{g}_t$, where $\text{comp} \colon \mathbb{R}^d \to \mathbb{R}^d$ generates a compression of the gradient, such as by lossy quantization or sparsification. We discuss different schemes below. However, too aggressive

compression can hurt the performance, unless it is implemented in a clever way: 1Bit-SGD [33, 37] combines gradient quantization with an error compensation technique, which is a memory or feedback mechanism. We in this work leverage this key mechanism but apply it within the more general setting of SGD. We will now sketch how the algorithm uses feedback to correct for errors accumulated in previous iterations. Roughly speaking, the method keeps track of a memory vector $\mathbf{m}$ which contains the sum of the information that has been suppressed thus far, i.e. $\mathbf{m}_{t+1} := \mathbf{m}_t + \mathbf{g}_t - \text{comp}(\mathbf{g}_t)$, and injects this information back in the next iteration, by transmitting $\text{comp}(\mathbf{m}_{t+1} + \mathbf{g}_{t+1})$ instead of only $\text{comp}(\mathbf{g}_{t+1})$. Note that updates of this kind are not unbiased (even if $\text{comp}(\mathbf{g}_{t+1})$ would be) and there is also no control over the delay after which the single coordinates are applied. These are some (technical) reasons why there exists no theoretical analysis of this scheme up to now.

In this paper we give a concise convergence rate analysis for SGD with memory and $k$-compression operators[2], such as (but not limited to) top-$k$ sparsification. Our analysis also supports ultra-sparsification operators for which $k < 1$, i.e. where *less than one* coordinate of the stochastic gradient is applied on average in (1). We not only provide the first convergence result of this method, but the result also shows that the method converges *at the same rate* as vanilla SGD.

## 1.1 Related Work

There are several ways to reduce the communication in SGD. For instance by simply increasing the amount of computation before communication, i.e. by using large mini-batches (see e.g. [12, 43]), or by designing communication-efficient schemes [45]. These approaches are a bit orthogonal to the methods we consider in this paper, which focus on quantization or sparsification of the gradient.

Several papers consider approaches that limit the number of bits to represent floating point numbers [13, 24, 31]. Recent work proposes adaptive tuning of the compression ratio [7]. Unbiased quantization operators not only limit the number of bits, but quantize the stochastic gradients in such a way that they are still unbiased estimators of the gradient [3, 41]. The ZipML framework also applies this technique to the data [44]. Sparsification methods reduce the number of non-zero entries in the stochastic gradient [3, 40].

A very aggressive sparsification method is to keep only very few coordinates of the stochastic gradient by considering only the coordinates with the largest magnitudes [1, 9]. In contrast to the unbiased schemes it is clear that such methods can only work by using some kind of error accumulation or feedback procedure, similar to the one the we have already discussed [33, 37], as otherwise certain coordinates could simply never be updated. However, in certain applications no feedback mechanism is needed [38]. Also more elaborate sparsification schemes have been introduced [21].

Asynchronous updates provide an alternative solution to disguise the communication overhead to a certain amount [19]. However, those methods usually rely on a sparsity assumption on the updates [25, 31], which is not realistic e.g. in deep learning. We like to advocate that combining gradient sparsification with those asynchronous schemes seems to be a promising approach, as it combines the best of both worlds. Other scenarios that could profit from sparsification are heterogeneous systems or specialized hardware, e.g. accelerators [11, 44].

Convergence proofs for SGD [29] typically rely on averaging the iterates [23, 27, 30], though convergence of the last iterate can also be proven [34]. For our convergence proof we rely on averaging techniques that give more weight to more recent iterates [17, 28, 34], as well as the perturbed iterate framework from Mania et al. [22] and techniques from [18, 36].

Simultaneous to our work, [4, 39] at NeurIPS 2018 propose related schemes. Whilst Tang et al. [39] only consider unbiased stochastic compression schemes, Alistarh et al. [4] study biased top-$k$ sparsification. Their scheme also uses a memory vector to compensate for the errors, but their analysis suffers from a slowdown proportional to $k$, which we can avoid here. Another simultaneous analysis of Wu et al. [42] at ICML 2018 is restricted to unbiased gradient compression. This scheme also critically relies on an error compensation technique, but in contrast to our work the analysis is restricted to quadratic functions and the scheme introduces two additional hyperparameters that control the feedback mechanism.

## 1.2 Contributions

We consider finite-sum convex optimization problems $f \colon \mathbb{R}^d \to \mathbb{R}$ of the form

$$f(\mathbf{x}) = \frac{1}{n} \sum_{i=1}^{n} f_i(\mathbf{x}), \qquad \mathbf{x}^\star := \underset{\mathbf{x} \in \mathbb{R}^d}{\arg\min} \, f(\mathbf{x}), \qquad f^\star := f(\mathbf{x}^\star), \qquad (2)$$

where each $f_i$ is $L$-smooth[3] and $f$ is $\mu$-strongly convex[4]. We consider a sequential sparsified SGD algorithm with error accumulation technique and prove convergence for $k$-compression operators, $0 < k \leq d$ (for instance the sparsification operators top-$k$ or random-$k$). For appropriately chosen stepsizes and an averaged iterate $\bar{\mathbf{x}}_T$ after $T$ steps we show convergence

$$\mathbb{E} \, f(\bar{\mathbf{x}}_T) - f^\star = \mathcal{O}\left(\frac{G^2}{\mu T}\right) + \mathcal{O}\left(\frac{\frac{d^2}{k^2} G^2 \kappa}{\mu T^2}\right) + \mathcal{O}\left(\frac{\frac{d^3}{k^3} G^2}{\mu T^3}\right), \qquad (3)$$

for $\kappa = \frac{L}{\mu}$ and $G^2 \geq \mathbb{E} \|\nabla f_i(\mathbf{x}_t)\|^2$. Not only is this, to the best of our knowledge, the first convergence result for sparsified SGD with memory, but the result also shows that the leading term $\mathcal{O}\left(\frac{G^2}{\mu T}\right)$ in the convergence rate is the same term as in the convergence rate as for vanilla SGD.

We introduce the method formally in Section 2 and show a sketch of the convergence proof in Section 3. In Section 4 we include a few numerical experiments for illustrative purposes. The experiments highlight that top-$k$ sparsification yields a very effective compression method and does not hurt convergence. We also report results for a parallel multi-core implementation of SGD with memory that show that the algorithm scales as well as asynchronous SGD and drastically decreases the communication cost without sacrificing the rate of convergence. We like to stress that the effectiveness of SGD variants with sparsification techniques has already been demonstrated in practice [1, 9, 21, 33, 37].

Although we do not yet provide convergence guarantees for parallel and asynchronous variants of the scheme, this is the main application of this method. For instance, we like to highlight that asynchronous SGD schemes [2, 25] could profit from the gradient sparsification. To demonstrate this use-case, we include in Section 4 a set of experiments for a multi-core implementation.

## 2 SGD with Memory

In this section we present the sparsified SGD algorithm with memory. First we introduce sparsification and quantization operators which allow us to drastically reduce the communication cost in comparison with vanilla SGD.

### 2.1 Compression and Sparsification Operators

We consider compression operators that satisfy the following contraction property:

**Definition 2.1** ($k$-contraction). *For a parameter $0 < k \leq d$, a $k$-contraction operator is a (possibly randomized) operator* $\mathrm{comp} \colon \mathbb{R}^d \to \mathbb{R}^d$ *that satisfies the contraction property*

$$\mathbb{E} \|\mathbf{x} - \mathrm{comp}(\mathbf{x})\|^2 \leq \left(1 - \frac{k}{d}\right) \|\mathbf{x}\|^2, \qquad \forall \mathbf{x} \in \mathbb{R}^d. \qquad (4)$$

The contraction property is sufficient to obtain all mathematical results that are derived in this paper. However, note that (4) does not imply that $\mathrm{comp}(\mathbf{x})$ is a necessarily sparse vector. Also dense vectors can satisfy (4). One of the main goals of this work is to derive communication efficient schemes, thus we are particularly interested in operators that also ensure that $\mathrm{comp}(\mathbf{x})$ can be encoded much more efficiently than the original $\mathbf{x}$.

The following two operators are examples of $k$-contraction operators with the additional property of being $k$-sparse vectors:

**Definition 2.2.** *For a parameter* $1 \leq k \leq d$, *the operators* $\mathrm{top}_k \colon \mathbb{R}^d \to \mathbb{R}^d$ *and* $\mathrm{rand}_k \colon \mathbb{R}^d \times \Omega_k \to \mathbb{R}^d$, *where* $\Omega_k = \binom{[d]}{k}$ *denotes the set of all $k$ element subsets of $[d]$, are defined for $\mathbf{x} \in \mathbb{R}^d$ as*

$$(\mathrm{top}_k(\mathbf{x}))_i := \begin{cases} (\mathbf{x})_{\pi(i)}, & \text{if } i \leq k, \\ 0 & \text{otherwise}, \end{cases} \qquad (\mathrm{rand}_k(\mathbf{x}, \omega))_i := \begin{cases} (\mathbf{x})_i, & \text{if } i \in \omega, \\ 0 & \text{otherwise}, \end{cases} \qquad (5)$$

*where $\pi$ is a permutation of $[d]$ such that $(|\mathbf{x}|)_{\pi(i)} \geq (|\mathbf{x}|)_{\pi(i+1)}$ for $i = 1, \ldots, d-1$. We abbreviate* $\mathrm{rand}_k(\mathbf{x})$ *whenever the second argument is chosen uniformly at random, $\omega \sim_{\mathrm{u.a.r.}} \Omega_k$.*

It is easy to see that both operators satisfy Definition 2.1 of being a $k$-contraction. For completeness the proof is included in Appendix A.1.

We note that our setting is more general than simply measuring sparsity in terms of the cardinality, i.e. the non-zero elements of vectors in $\mathbb{R}^d$. Instead, Definition 2.1 can also be considered for quantization or e.g. floating point representation of each entry of the vector. In this setting we would for instance measure sparsity in terms of the number of bits that are needed to encode the vector. By this, we can also use stochastic rounding operators (similar as the ones used in [3], but with different scaling) as compression operators according to (4). Also gradient dropping [1] trivially satisfies (4), though with different parameter $k$ in each iteration.

**Remark 2.3** (Ultra-sparsification). *We like to highlight that many other operators do satisfy Definition 2.1, not only the two examples given in Definition 2.2. As a notable variant is to pick a random coordinate of a vector with probability $\frac{k}{d}$, for $0 < k \leq 1$, property (4) holds even if $k < 1$. I.e. it suffices to transmit on average less than one coordinate per iteration (this would then correspond to a mini-batch update).*

## 2.2 Variance Blow-up for Unbiased Updates

Before introducing SGD with memory we first discuss a motivating example. Consider the following variant of SGD, where $(d - k)$ random coordinates of the stochastic gradient are dropped:

$$\mathbf{x}_{t+1} := \mathbf{x}_t - \eta_t \mathbf{g}_t, \qquad\qquad \mathbf{g}_t := \tfrac{d}{k} \cdot \mathrm{rand}_k(\nabla f_i(\mathbf{x}_t)), \qquad (6)$$

where $i \sim_{\mathrm{u.a.r}} [n]$. It is important to note that the update is unbiased, i.e. $\mathbb{E}\, \mathbf{g}_t = \nabla f(\mathbf{x})$. For carefully chosen stepsizes $\eta_t$ this algorithm converges at rate $\mathcal{O}\!\left(\frac{\sigma^2}{t}\right)$ on strongly convex and smooth functions $f$, where $\sigma^2$ is an upper bound on the variance, see for instance [46]. We have

$$\sigma^2 = \mathbb{E}\left\| \tfrac{d}{k}\,\mathrm{rand}_k(\nabla f_i(\mathbf{x})) - \nabla f(\mathbf{x}) \right\|^2 \leq \mathbb{E}\left\| \tfrac{d}{k}\,\mathrm{rand}_k(\nabla f_i(\mathbf{x})) \right\|^2 \leq \tfrac{d}{k}\, \mathbb{E}_i \left\| \nabla f_i(\mathbf{x}) \right\|^2 \leq \tfrac{d}{k} G^2$$

where we used the variance decomposition $\mathbb{E} \left\| X - \mathbb{E}\, X \right\|^2 = \mathbb{E} \left\| X \right\|^2 - \left\| \mathbb{E}\, X \right\|^2$ and the standard assumption $\mathbb{E}_i \left\| \nabla f_i(\mathbf{x}) \right\|^2 \leq G^2$. Hence, when $k$ is small this algorithm requires $d$ times more iterations to achieve the same error guarantee as vanilla SGD with $k = d$.

It is well known that by using mini-batches the variance of the gradient estimator can be reduced. If we consider in (6) the estimator $\mathbf{g}_t := \tfrac{d}{k} \cdot \mathrm{rand}_k\!\left(\tfrac{1}{\tau} \sum_{i \in \mathcal{I}_\tau} \nabla f_i(\mathbf{x}_t)\right)$ for $\tau = \lceil \tfrac{k}{d} \rceil$, and $\mathcal{I}_\tau \sim_{\mathrm{u.a.r.}} \binom{[n]}{k}$ instead, we have

$$\sigma^2 = \mathbb{E} \left\| \mathbf{g}_t - \nabla f(\mathbf{x}_t) \right\|^2 \leq \mathbb{E} \left\| \tfrac{d}{k} \cdot \mathrm{rand}_k\!\left(\tfrac{1}{\tau} \sum_{i \in \mathcal{I}_\tau} \nabla f_i(\mathbf{x}_t)\right) \right\|^2 \leq \tfrac{d}{k\tau}\, \mathbb{E}_i \left\| \nabla f_i(\mathbf{x}_t) \right\|^2 \leq G^2. \quad (7)$$

This shows that, when using mini-batches of appropriate size, the sparsification of the gradient does not hurt convergence. However, by increasing the mini-batch size, we increase the computation by a factor of $\frac{d}{k}$.

These two observations seem to indicate that the factor $\frac{d}{k}$ is inevitably lost, either by increased number of iterations or increased computation. However, this is no longer true when the information in (6) is not dropped, but kept in memory. To illustrate this, assume $k = 1$ and that index $i$ has not been selected by the $\mathrm{rand}_1$ operator in iterations $t = t_0, \cdots, t_{s-1}$, but is selected in iteration $t_s$. Then the memory $\mathbf{m}_{t_s} \in \mathbb{R}^d$ contains this past information $(\mathbf{m}_{t_s})_i = \sum_{t=t_0}^{t_{s-1}} (\nabla f_{i_t}(\mathbf{x}_t))_i$. Intuitively, we would expect that the variance of this estimator is now reduced by a factor of $s$ compared to the naïve estimator in (6), similar to the mini-batch update in (7). Indeed, SGD with memory converges at the same rate as vanilla SGD, as we will demonstrate below.

| **Algorithm 1** MEM-SGD | **Algorithm 2** PARALLEL-MEM-SGD |
|---|---|
| 1: Initialize variables $\mathbf{x}_0$ and $\mathbf{m}_0 = \mathbf{0}$ | 1: Initialize shared variable $\mathbf{x}$ |
| 2: **for** $t$ **in** $0 \ldots T - 1$ **do** |    and $\mathbf{m}_0^w = \mathbf{0}, \forall w \in [W]$ |
| 3:    Sample $i_t$ uniformly in $[n]$ | 2: **parallel for** $w$ **in** $1 \ldots W$ **do** |
| 4:    $\mathbf{g}_t \leftarrow \mathrm{comp}_k(\mathbf{m}_t + \eta_t \nabla f_{i_t}(\mathbf{x}_t))$ | 3:    **for** $t$ **in** $0 \ldots T - 1$ **do** |
| 5:    $\mathbf{x}_{t+1} \leftarrow \mathbf{x}_t - \mathbf{g}_t$ | 4:      Sample $i_t^w$ uniformly in $[n]$ |
| 6:    $\mathbf{m}_{t+1} \leftarrow \mathbf{m}_t + \eta_t \nabla f_{i_t}(\mathbf{x}_t) - \mathbf{g}_t$ | 5:      $\mathbf{g}_t^w \leftarrow \mathrm{comp}_k(\mathbf{m}_t^w + \eta_t \nabla f_{i_t^w}(\mathbf{x}))$ |
| 7: **end for** | 6:      $\mathbf{x} \leftarrow \mathbf{x} - \mathbf{g}_t^w$   ▷ shared memory |
|  | 7:      $\mathbf{m}_{t+1}^w \leftarrow \mathbf{m}_t^w + \eta_t \nabla f_{i_t^w}(\mathbf{x}) - \mathbf{g}_t^w$ |
|  | 8:    **end for** |
|  | 9: **end parallel for** |

Figure 1: *Left:* The MEM-SGD algorithm. *Right:* Implementation for multi-core experiments.

## 2.3 SGD with Memory: Algorithm and Convergence Results

We consider the following algorithm for parameter $0 < k \leq d$, using a compression operator $\mathrm{comp}_k \colon \mathbb{R}^d \to \mathbb{R}^d$ which is a $k$-contraction (Definition 2.1)

$$\mathbf{x}_{t+1} := \mathbf{x}_t - \mathbf{g}_t\,, \quad \mathbf{g}_t := \mathrm{comp}_k(\mathbf{m}_t + \eta_t \nabla f_{i_t}(\mathbf{x}_t))\,, \quad \mathbf{m}_{t+1} := \mathbf{m}_t + \eta_t \nabla f_{i_t}(\mathbf{x}_t) - \mathbf{g}_t\,, \quad (8)$$

where $i_t \sim_{\mathrm{u.a.r.}} [n]$, $\mathbf{m}_0 := 0$ and $\{\eta_t\}_{t \geq 0}$ denotes a sequence of stepsizes. The pseudocode is given in Algorithm 1. Note that the gradients get multiplied with the stepsize $\eta_t$ at the timestep $t$ when they put into memory, and not when they are (partially) retrieved from the memory.

We state the precise convergence result for Algorithm 1 in Theorem 2.4 below. In Remark 2.6 we give a simplified statement in big-$\mathcal{O}$ notation for a specific choice of the stepsizes $\eta_t$.

**Theorem 2.4.** *Let $f_i$ be $L$-smooth, $f$ be $\mu$-strongly convex, $0 < k \leq d$, $\mathbb{E}_i \|\nabla f_i(\mathbf{x}_t)\|^2 \leq G^2$ for $t = 0, \ldots, T - 1$, where $\{\mathbf{x}_t\}_{t \geq 0}$ are generated according to (8) for stepsizes $\eta_t = \frac{8}{\mu(a+t)}$ and shift parameter $a > 1$. Then for $\alpha > 4$ such that $\frac{(\alpha+1)\frac{d}{k} + \rho}{\rho+1} \leq a$, with $\rho := \frac{4\alpha}{(\alpha-4)(\alpha+1)^2}$, it holds*

$$\mathbb{E} f(\bar{\mathbf{x}}_T) - f^\star \leq \frac{4T(T + 2a)}{\mu S_T} G^2 + \frac{\mu a^3}{8 S_T} \|\mathbf{x}_0 - \mathbf{x}^\star\|^2 + \frac{64 T \left(1 + 2\frac{L}{\mu}\right)}{\mu S_T} \left(\frac{4\alpha}{\alpha - 4}\right) \frac{d^2}{k^2} G^2\,, \quad (9)$$

*where $\bar{\mathbf{x}}_T = \frac{1}{S_T} \sum_{t=0}^{T-1} w_t \mathbf{x}_t$, for $w_t = (a + t)^2$, and $S_T = \sum_{t=0}^{T-1} w_t \geq \frac{1}{3} T^3$.*

**Remark 2.5** (Choice of the shift $a$)**.** *Theorem 2.4 says that for any shift $a > 1$ there is a parameter $\alpha(a) > 4$ such that (9) holds. However, for the choice $a = \mathcal{O}(1)$ one has to set $\alpha$ such that $\frac{\alpha}{\alpha-4} = \Omega(\frac{d}{k})$ and the last term in (9) will be of order $\mathcal{O}(\frac{d^3}{k^3 T^2})$, thus requiring $T = \Omega(\frac{d^{1.5}}{k^{1.5}})$ steps to yield convergence. For $\alpha \geq 5$ we have $\frac{\alpha}{\alpha-4} = \mathcal{O}(1)$ and the last term is only of order $\mathcal{O}(\frac{d^2}{k^2 T^2})$ instead. However, this requires typically a large shift. Observe $\frac{(\alpha+1)\frac{d}{k}+\rho}{\rho+1} \leq 1 + (\alpha+1)\frac{d}{k} \leq (\alpha+2)\frac{d}{k}$, i.e. setting $a = (\alpha + 2)\frac{d}{k}$ is enough. We like to stress that in general it is not advisable to set $a \gg (\alpha + 2)\frac{d}{k}$ as the first two terms in (9) depend on $a$. In practice, it often suffices to set $a = \frac{d}{k}$, as we will discuss in Section 4.*

**Remark 2.6.** *As discussed in Remark 2.5 above, setting $\alpha = 5$ and $a = (\alpha + 2)\frac{d}{k}$ is feasible. With this choice, equation (9) simplifies to*

$$\mathbb{E} f(\bar{\mathbf{x}}_T) - f^\star \leq \mathcal{O}\left(\frac{G^2}{\mu T}\right) + \mathcal{O}\left(\frac{\frac{d^2}{k^2} G^2 \kappa}{\mu T^2}\right) + \mathcal{O}\left(\frac{\frac{d^3}{k^3} G^2}{\mu T^3}\right)\,, \quad (10)$$

*for $\kappa = \frac{L}{\mu}$. To estimate the second term in (9) we used the property $\mathbb{E}\, \mu \|\mathbf{x}_0 - \mathbf{x}^\star\| \leq 2G$ for $\mu$-strongly convex $f$, as derived in [28, Lemma 2]. We observe that for large $T$ the first term, $\mathcal{O}(\frac{G^2}{\mu T})$, is dominating the rate. This is the same term as in the convergence rate of vanilla SGD [17].*

## 3 Proof Outline

We now give an outline of the proof. The proofs of the lemmas are given in Appendix A.2.

**Perturbed iterate analysis.** Inspired by the perturbed iterate framework in [22] and [18] we first define a virtual sequence $\{\tilde{\mathbf{x}}_t\}_{t \geq 0}$ in the following way:

$$\tilde{\mathbf{x}}_0 = \mathbf{x}_0\,, \qquad\qquad\qquad \tilde{\mathbf{x}}_{t+1} = \tilde{\mathbf{x}}_t - \eta_t \nabla f_{i_t}(\mathbf{x}_t)\,, \qquad (11)$$

where the sequences $\{\mathbf{x}_t\}_{t \geq 0}$, $\{\eta_t\}_{t \geq 0}$ and $\{i_t\}_{t \geq 0}$ are the same as in (8). Notice that

$$\tilde{\mathbf{x}}_t - \mathbf{x}_t = \left(\mathbf{x}_0 - \sum_{j=0}^{t-1} \eta_j \nabla f_{i_j}(\mathbf{x}_j)\right) - \left(\mathbf{x}_0 - \sum_{j=0}^{t-1} \mathbf{g}_j\right) = \mathbf{m}_t\,. \qquad (12)$$

**Lemma 3.1.** *Let $\{\mathbf{x}_t\}_{t \geq 0}$ and $\{\tilde{\mathbf{x}}_t\}_{t \geq 0}$ be defines as in* (8) *and* (11) *and let $f_i$ be $L$-smooth and $f$ be $\mu$-strongly convex with $\mathbb{E}_i \|\nabla f_i(\mathbf{x}_t)\|^2 \leq G^2$. Then*

$$\mathbb{E} \|\tilde{\mathbf{x}}_{t+1} - \mathbf{x}^\star\|^2 \leq \left(1 - \frac{\eta_t \mu}{2}\right) \mathbb{E} \|\tilde{\mathbf{x}}_t - \mathbf{x}^\star\|^2 + \eta_t^2 G^2 - \eta_t e_t + \eta_t(\mu + 2L) \mathbb{E} \|\mathbf{m}_t\|^2\,, \qquad (13)$$

*where $e_t := \mathbb{E} f(\mathbf{x}_t) - f^\star$.*

**Bounding the memory.** From equation (13) it becomes clear that we should derive an upper bound on $\mathbb{E} \|\mathbf{m}_t\|^2$. For this we will use the contraction property (4) of the compression operators.

**Lemma 3.2.** *Let $\{\mathbf{x}_t\}_{t \geq 0}$ as defined in* (8) *for $0 < k \leq d$, $\mathbb{E}_i \|\nabla f_i(\mathbf{x}_t)\|^2 \leq G^2$ and stepsizes $\eta_t = \frac{8}{\mu(a+t)}$ with $a, \alpha > 4$, as in Theorem 2.4. Then*

$$\mathbb{E} \|\mathbf{m}_t\|^2 \leq \eta_t^2 \frac{4\alpha}{\alpha - 4} \frac{d^2}{k^2} G^2\,. \qquad (14)$$

**Optimal averaging.** Similar as discussed in [17, 28, 34] we have to define a suitable averaging scheme for the iterates $\{\mathbf{x}_t\}_{t \geq 0}$ to get the optimal convergence rate. In contrast to [17] that use linearly increasing weights, we use quadratically increasing weights, as for instance [34, 36].

**Lemma 3.3.** *Let $\{a_t\}_{t \geq 0}$, $a_t \geq 0$, $\{e_t\}_{t \geq 0}$, $e_t \geq 0$, be sequences satisfying*

$$a_{t+1} \leq \left(1 - \frac{\mu \eta_t}{2}\right) a_t + \eta_t^2 A + \eta_t^3 B - \eta_t e_t\,, \qquad (15)$$

*for $\eta_t = \frac{8}{\mu(a+t)}$ and constants $A, B \geq 0$, $\mu > 0$, $a > 1$. Then*

$$\frac{1}{S_T} \sum_{t=0}^{T-1} w_t e_t \leq \frac{\mu a^3}{8 S_T} a_0 + \frac{4T(T + 2a)}{\mu S_T} A + \frac{64T}{\mu^2 S_T} B\,, \qquad (16)$$

*for $w_t = (a+t)^2$ and $S_T := \sum_{t=0}^{T-1} w_t = \frac{T}{6}\left(2T^2 + 6aT - 3T + 6a^2 - 6a + 1\right) \geq \frac{1}{3}T^3$.*

**Proof of Theorem 2.4.** The proof of the theorem immediately follows from the three lemmas that we have presented in this section and convexity of $f$, i.e. we have $\mathbb{E} f(\bar{\mathbf{x}}_T) - f^\star \leq \frac{1}{S_T} \sum_{t=0}^{T-1} w_t e_t$ in (16), for constants $A = G^2$ and $B = (\mu + 2L)\frac{4\alpha}{\alpha - 4}\frac{d^2}{k^2} G^2$. □

# 4 Experiments

We present numerical experiments to illustrate the excellent convergence properties and communication efficiency of MEM-SGD. As the usefulness of SGD with sparsification techniques has already been shown in practical applications [1, 9, 21, 33, 37] we focus here on a few particular aspects. First, we verify the impact of the initial learning rate that did come up in the statement of Theorem 2.4. We then compare our method with QSGD [3] which decreases the communication cost in SGD by using random quantization operators, but without memory. Finally, we show the performance of the parallel SGD depicted in Algorithm 2 in a multi-core setting with shared memory and compare the speed-up to asynchronous SGD.

## 4.1 Experimental Setup

**Models.** The experiments focus on the performance of MEM-SGD applied to logistic regression. The associated objective function is $\frac{1}{n} \sum_{i=1}^n \log(1 + \exp(-b_i \mathbf{a}_i^\top \mathbf{x})) + \frac{\lambda}{2}\|\mathbf{x}\|^2$, where $\mathbf{a}_i \in \mathbb{R}^d$ and $b_i \in \{-1, +1\}$ are the data samples, and we employ a standard $L2$-regularizer. The regularization parameter is set to $\lambda = 1/n$ for both datasets following [32].

|          | $n$     | $d$    | density |
|----------|---------|--------|---------|
| **epsilon**  | 400'000 | 2'000  | 100%    |
| **RCV1-test** | 677'399 | 47'236 | 0.15%   |

Table 1: Datasets statistics.

|          | parameter | value  |
|----------|-----------|--------|
| **epsilon**  | $\gamma$  | 2      |
|          | $a$       | $d/k$  |
| **RCV1-test** | $\gamma$  | 2      |
|          | $a$       | $10d/k$ |

Table 2: Learning rate $\eta_t = \gamma/(\lambda(t+a))$.

**Datasets.** We consider a dense dataset, *epsilon* [35], as well as a sparse dataset, *RCV1* [20] where we train on the larger test set. Statistics on the datasets are listed in Table 1.

**Implementation.** We use Python3 and the numpy library [15]. Our code is open-source and publicly available at `github.com/epfml/sparsifiedSGD`. We emphasize that our high level implementation is not optimized for speed per iteration but for readability and simplicity. We only report convergence per iteration and relative speedups, but not wall-clock time because unequal efforts have been made to speed up the different implementations. Plots additionally show the baseline computed with the standard optimizer `LogisticSGD` of scikit-learn [26]. Experiments were run on an Ubuntu 18.04 machine with a 24 cores processor Intel® Xeon® CPU E5-2680 v3 @ 2.50GHz.

## 4.2 Verifying the Theory

We study the convergence of the method using the stepsizes $\eta_t = \gamma/(\lambda(t+a))$ and hyperparameters $\gamma$ and $a$ set as in Table 2. We compute the final estimate $\bar{\mathbf{x}}$ as a weighted average of all iterates $\mathbf{x}_t$ with weights $w_t = (t+a)^2$ as indicated by Theorem 2.4. The results are depicted in Figure 2. We use $k \in \{1, 2, 3\}$ for *epsilon* and $k \in \{10, 20, 30\}$ for *RCV1* to increase the difference with large number of features. The $\text{top}_k$ variant consistently outperforms $\text{rand}_k$ and sometimes outperforms vanilla SGD, which is surprising and might come from feature characteristics of the datasets. We also evaluate the impact of the delay $a$ in the learning rate: setting it to 1 instead of order $\mathcal{O}(d/k)$ dramatically hurts the memory and requires time to recover from the high initial learning rate (labeled "without delay" in Figure 2).

We experimentally verified the convergence properties of MEM-SGD for different sparsification operators and stepsizes but we want to further evaluate its fundamental benefits in terms of sparsity enforcement and reduction of the communication bottleneck. The gain in communication cost of SGD with memory is very high for dense datasets—using the $\text{top}_1$ strategy on *epsilon* dataset improves the amount of communication by $10^3$ compared to SGD. For the sparse dataset, SGD can readily use the given sparsity of the gradients. Nevertheless, the improvement for $\text{top}_{10}$ on *RCV1* is of approximately an order of magnitude.

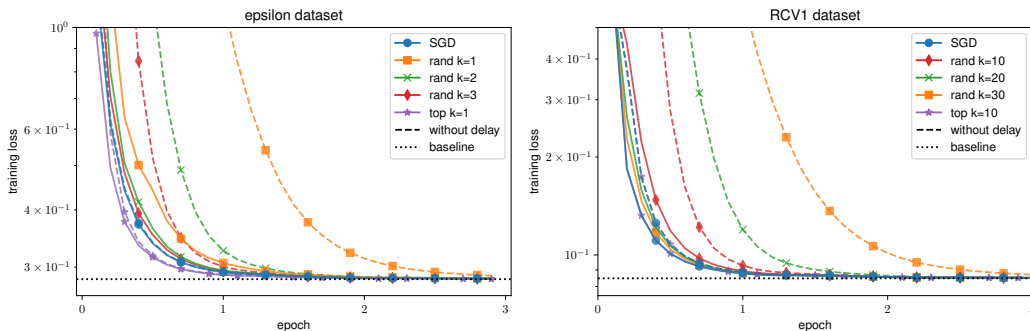

Figure 2: Convergence of MEM-SGD using different sparsification operators compared to full SGD with theoretical learning rates (parameters in Table 2).

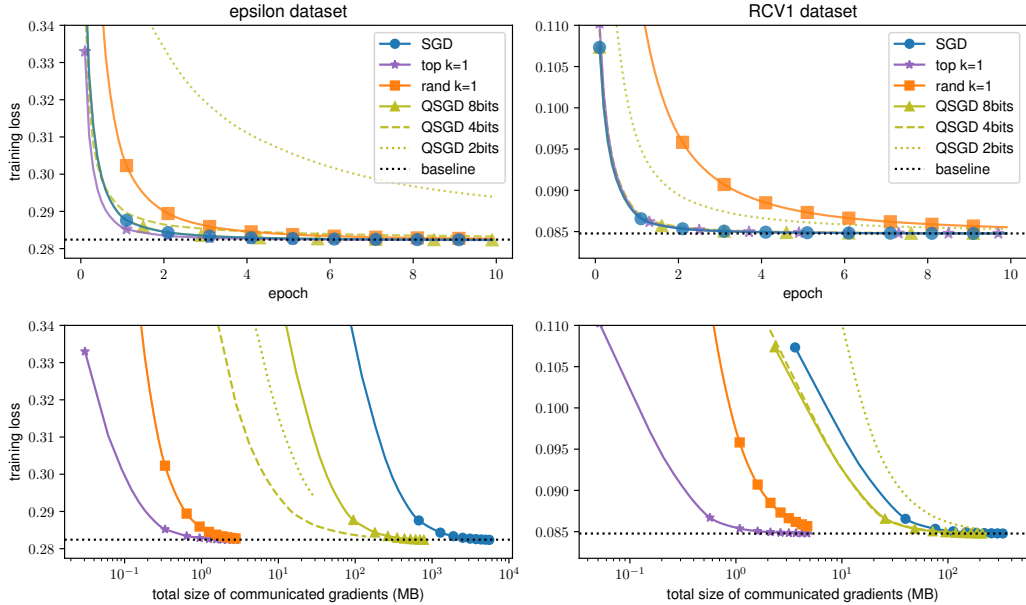

Figure 3: MEM-SGD and QSGD convergence comparison. *Top row:* convergence in number of iterations. *Bottom row:* cumulated size of the communicated gradients during training. We compute the loss 10 times per epoch and remove the point at 0MB for clarity.

## 4.3 Comparison with QSGD

Now we compare MEM-SGD with the QSGD compression scheme [3] which reduces communication cost by random quantization. The accuracy (and the compression ratio) in QSGD is controlled by a parameter $s$, corresponding to the number of quantization levels. Ideally, we would like to set the quantization precision in QSGD such that the number of bits transmitted by QSGD and MEM-SGD are identical and compare their convergence properties. However, even for the lowest precision, QSGD needs to send the sign and index of $\mathcal{O}(\sqrt{d})$ coordinates. It is therefore not possible to reach the compression level of sparsification operators such as top-$k$ or random-$k$, that only transmit a constant number of bits per iteration (up to logarithmic factors).[5] Hence, we did not enforce this condition and resorted to pick reasonable levels of quantization in QSGD ($s = 2^b$ with $b \in \{2, 4, 8\}$). Note that $b$-bits stands for the number of bits used to encode $s = 2^b$ levels but the number of bits transmitted in QSGD can be reduced using Elias coding. As a fair comparison in practice, we chose a standard learning rate $\gamma_0/(1 + \gamma_0 \lambda t)^{-1}$ [6], tuned the hyperparameter $\gamma_0$ on a subset of each dataset (see Appendix B). Figure 3 shows that MEM-SGD with $\text{top}_1$ on *epsilon* and *RCV1* converges as fast as QSGD in term of iterations for 8 and 4-bits. As shown in the bottom of Figure 3, we are transmitting two orders of magnitude fewer bits with the $\text{top}_1$ sparsifier concluding that sparsification offers a much more aggressive and performant strategy than quantization.

## 4.4 Multicore experiment

We implement a parallelized version of MEM-SGD, as depicted in Algorithm 2. The enforced sparsity allows us to do the update in shared memory using a lock-free mechanism as in [25]. For this experiment we evaluate the final iterate $\mathbf{x}_T$ instead of the weighted average $\bar{\mathbf{x}}_T$ above, and use the learning rate $\eta_t \equiv (1 + t)^{-1}$.

Figure 4 shows the speed-up obtained when increasing the number of cores. We see that both sparsified SGD and vanilla SGD have a linear speed-up, the slopes are dependent of the implementation details. But we observe that PARALLEL-MEM-SGD with a reasonable sparsification parameter $k$ does not suffer of having multiple independent memories. The experiment is run on a single machine with a

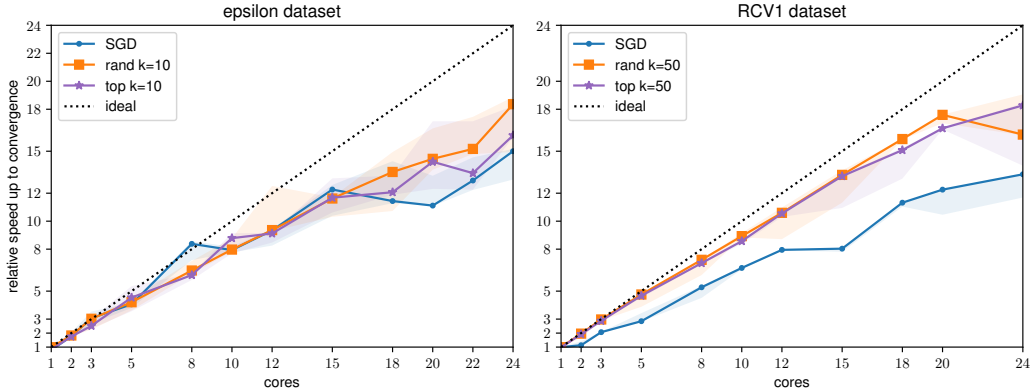

Figure 4: Multicore wall-clock time speed up comparison between MEM-SGD and lock-free SGD. The colored area depicts the best and worst results of 3 independent runs for each dataset.

24 core processor, hence no inter-node communication is used. The main advantage of our method— overcoming the communication bottleneck— would be even more visible in a multi-node setup. In this asynchronous setup, SGD with memory computes gradients on stale iterates that differ only by a few coordinates. It encounters fewer inconsistent read/write operations than lock free asynchronous SGD and exhibits better scaling properties on the RCV1 dataset. The $\text{top}_k$ operator performs better than $\text{rand}_k$ in the sequential setup, but this is not the case in the parallel setup.

## 5    Conclusion

We provide the first concise convergence analysis of sparsified SGD [1, 9, 33, 37]. This extremely communication-efficient variant of SGD enforces sparsity of the applied updates by only updating a constant number of coordinates in every iteration. This way, the method overcomes the communication bottleneck of SGD, while still enjoying the same convergence rate in terms of stochastic gradient computations.

Our experiments verify the drastic reduction in communication cost by demonstrating that MEM-SGD requires one to two orders of magnitude less bits to be communicated than QSGD [3] while converging to the same accuracy. The experiments show an advantage for the top-$k$ sparsification over random sparsification in the serial setting, but not in the multi-core shared memory implementation. There, both schemes are on par, and show better scaling than a simple shared memory implementation that just writes the unquantized updates in a lock-free asynchronous fashion (like Hogwild! [25]).

The theoretical insights to MEM-SGD that were developed here should facilitate the analysis of the same scheme in the parallel (as developped in [8]) and the distributed setting. It has already been shown in practice that gradient sparsification can be efficiently applied to bandwidth memory limited systems such as multi-GPU training for neural networks [1, 9, 21, 33, 37]. By delivering sparsity no matter if the original gradients were sparse or not, our scheme is not only communication efficient, but becomes more eligible for asynchronous implementations as well. While those were so far limited by strict sparsity assumptions (as e.g. in [25]), our approach might make such methods much more widely applicable.

## Acknowledgments

We would like to thank Dan Alistarh for insightful discussions in the early stages of this project and Frederik Künstner for his useful comments on the various drafts of this manuscript. We acknowledge funding from SNSF grant 200021_175796, Microsoft Research JRC project 'Coltrain', as well as a Google Focused Research Award.

## Footnotes

[1] Note that the stochastic gradients $\mathbf{g}_t$ are dense vectors for the setting of training neural networks. The $\mathbf{g}_t$ themselves can be sparse for generalized linear models under the additional assumption that the data is sparse.

[2]See Definition 2.1.

[3]$f_i(\mathbf{y}) \leq f_i(\mathbf{x}) + \langle \nabla f_i(\mathbf{x}), \mathbf{y} - \mathbf{x} \rangle + \frac{L}{2} \|\mathbf{y} - \mathbf{x}\|^2, \forall \mathbf{x}, \mathbf{y} \in \mathbb{R}^d, i \in [n].$

[4]$f(\mathbf{y}) \geq f(\mathbf{x}) + \langle \nabla f(\mathbf{x}), \mathbf{y} - \mathbf{x} \rangle + \frac{\mu}{2} \|\mathbf{y} - \mathbf{x}\|^2, \forall \mathbf{x}, \mathbf{y} \in \mathbb{R}^d.$

[5]Encoding the indices of the top-$k$ or random-$k$ elements can be done with additional $\mathcal{O}(k \log d)$ bits. Note that $\log d \le 32 \le \sqrt{d}$ for both our examples.

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
