[Supplementary Material]

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

# Appendix

## A   Proofs

### A.1   Useful facts

**Lemma A.1.** *For $\mathbf{x} \in \mathbb{R}^d$, $1 \leq k \leq d$, and operator $\mathrm{comp}_k \in \{\mathrm{top}_k, \mathrm{rand}_k\}$ it holds*

$$\mathbb{E}\left\|\mathrm{comp}_k(\mathbf{x}) - \mathbf{x}\right\|^2 \leq \left(1 - \frac{k}{d}\right)\|\mathbf{x}\|^2 . \tag{17}$$

*Proof.* From the definition of the operators, for all $\mathbf{x}$ in $\mathbb{R}^d$ we have

$$\left\|\mathbf{x} - \mathrm{top}_k(\mathbf{x})\right\|^2 \leq \left\|\mathbf{x} - \mathrm{rand}_k(\mathbf{x})\right\|^2 \tag{18}$$

and we apply the expectation

$$\mathbb{E}_\omega \left\|\mathbf{x} - \mathrm{rand}_k(\mathbf{x})\right\|^2 = \frac{1}{|\Omega_k|} \sum_{\omega \in \Omega_k} \sum_{i=1}^d \mathbf{x}_i^2 \mathbb{I}\{i \notin \omega\} = \sum_{i=1}^d x_i^2 \sum_{\omega \in \Omega_k} \frac{\mathbb{I}\{i \notin \omega\}}{|\Omega_k|} = \left(1 - \frac{k}{d}\right)\|\mathbf{x}\|^2 \tag{19}$$

which concludes the proof. $\square$

**Lemma A.2.** *Let $\eta_t = \frac{1}{c+t}$, for $c \geq 1$. Then $\eta_t^2 \left(1 - \frac{2}{c}\right) \leq \eta_{t+1}^2$.*

*Proof.* Observe

$$\eta_t^2 \left(1 - \frac{2}{c}\right) = \frac{c-2}{c(c+t)^2} \leq \frac{c-2}{(c+t+1)^2(c-2)} = \eta_{t+1}^2 . \tag{20}$$

where the inequality follows from

$$(c+t+1)^2(c-2) = c(c+t)^2 + \underbrace{(c-2)(1+2(t+c)) - 2(c+t)^2}_{=-2t^2 - 2ct - 4t - 3c - 2 \leq 0} \tag{21}$$

$\square$

### A.2   Proof of the Main Theorem

*Proof of Lemma 3.1.* Using the update equation (11) we have

$$\left\|\tilde{\mathbf{x}}_{t+1} - \mathbf{x}^\star\right\|^2 = \left\|\tilde{\mathbf{x}}_t - \mathbf{x}^\star\right\|^2 + \eta_t^2 \left\|\nabla f_{i_t}(\mathbf{x}_t)\right\|^2 - 2\eta_t \left\langle \mathbf{x}_t - \mathbf{x}^\star, \nabla f_{i_t}(\mathbf{x}_t)\right\rangle + 2\eta_t \left\langle \mathbf{x}_t - \tilde{\mathbf{x}}_t, \nabla f_{i_t}(\mathbf{x}_t)\right\rangle . \tag{22}$$

And by applying expectation

$$\mathbb{E}_{i_t} \left\|\tilde{\mathbf{x}}_{t+1} - \mathbf{x}^\star\right\|^2 \leq \left\|\tilde{\mathbf{x}}_t - \mathbf{x}^\star\right\|^2 + \eta_t^2 G^2 - 2\eta_t \left\langle \mathbf{x}_t - \mathbf{x}^\star, \nabla f(\mathbf{x}_t)\right\rangle + 2\eta_t \left\langle \mathbf{x}_t - \tilde{\mathbf{x}}_t, \nabla f(\mathbf{x}_t)\right\rangle . \tag{23}$$

To upper bound the third term, we use the same estimates as in [18, Appendix C.3]: By strong convexity, $f(\mathbf{y}) \geq f(\mathbf{x}) + \left\langle \nabla f(\mathbf{x}), \mathbf{y} - \mathbf{x}\right\rangle + \frac{\mu}{2}\|\mathbf{y} - \mathbf{x}\|^2$ for $\mathbf{x}, \mathbf{y} \in \mathbb{R}^d$, hence

$$- \left\langle \mathbf{x}_t - \mathbf{x}^\star, \nabla f(\mathbf{x}_t)\right\rangle \leq - (f(\mathbf{x}_t) - f^\star) - \frac{\mu}{2}\left\|\mathbf{x}_t - \mathbf{x}^\star\right\|^2 \tag{24}$$

and with $\|\mathbf{a} + \mathbf{b}\|^2 \leq 2\|\mathbf{a}\|^2 + 2\|\mathbf{b}\|^2$ we further have

$$- \left\|\mathbf{x}_t - \mathbf{x}^\star\right\|^2 \leq \left\|\mathbf{x}_t - \tilde{\mathbf{x}}_t\right\|^2 - \frac{1}{2}\left\|\tilde{\mathbf{x}}_t - \mathbf{x}^\star\right\|^2 . \tag{25}$$

Putting these two estimates together, we can bound (23) as follows:

$$\mathbb{E}_{i_t} \left\|\tilde{\mathbf{x}}_{t+1} - \mathbf{x}^\star\right\|^2 \leq \left(1 - \frac{\eta_t \mu}{2}\right)\left\|\tilde{\mathbf{x}}_t - \mathbf{x}^\star\right\|^2 + \eta_t^2 G^2 - 2\eta_t e_t + \eta_t \mu \left\|\mathbf{x}_t - \tilde{\mathbf{x}}_t\right\|^2 + 2\eta_t \left\langle \mathbf{x}_t - \tilde{\mathbf{x}}_t, \nabla f(\mathbf{x}_t)\right\rangle , \tag{26}$$

where $e_t = \mathbb{E} f(\mathbf{x}_t) - f^\star$. We now estimate the last term. As each $f_i$ is $L$-smooth also $f$ is $L$-smooth, i.e. satisfies $f(\mathbf{x}) - f(\mathbf{y}) - \langle \nabla f(\mathbf{y}), \mathbf{x} - \mathbf{y} \rangle \geq \frac{1}{2L} \|\nabla f(\mathbf{y}) - \nabla f(\mathbf{x})\|^2$. Together with $2\langle a, b \rangle \leq \gamma \|a\|^2 + \gamma^{-1} \|b\|^2$ we have

$$\langle \mathbf{x}_t - \tilde{\mathbf{x}}_t, \nabla f(\mathbf{x}_t) \rangle \leq \frac{1}{2} \left( 2L \|\mathbf{x}_t - \tilde{\mathbf{x}}_t\|^2 + \frac{1}{2L} \|\nabla f(\mathbf{x}_t)\|^2 \right) \tag{27}$$

$$= L \|\mathbf{x}_t - \tilde{\mathbf{x}}_t\|^2 + \frac{1}{4L} \|\nabla f(\mathbf{x}_t) - \nabla f(\mathbf{x}^\star)\|^2 \tag{28}$$

$$\leq L \|\mathbf{x}_t - \tilde{\mathbf{x}}_t\|^2 + \frac{1}{2} \left( f(\mathbf{x}_t) - f^\star \right) . \tag{29}$$

Combining with (26) we have

$$\mathbb{E}_{i_t} \|\tilde{\mathbf{x}}_{t+1} - \mathbf{x}^\star\|^2 \leq \left( 1 - \frac{\eta_t \mu}{2} \right) \|\tilde{\mathbf{x}}_t - \mathbf{x}^\star\|^2 + \eta_t^2 G^2 - \eta_t e_t + \eta_t (\mu + 2L) \|\mathbf{x}_t - \tilde{\mathbf{x}}_t\|^2 , \tag{30}$$

and the claim follows with (12). $\qquad\square$

*Proof of Lemma 3.2.* First, observe that by Lemma A.1 and $\|\mathbf{a} + \mathbf{b}\|^2 \leq (1 + \gamma) \|\mathbf{a}\|^2 + (1 + \gamma^{-1}) \|\mathbf{b}\|^2$ for $\gamma > 0$ we have

$$\mathbb{E} \|\mathbf{m}_{t+1}\|^2 \leq \left( 1 - \frac{k}{d} \right) \|\mathbf{m}_t + \eta_t \nabla f_{i_t}(\mathbf{x}_t)\|^2 \tag{31}$$

$$\leq \left( 1 - \frac{k}{d} \right) \left( \left( 1 + \frac{k}{2d} \right) \mathbb{E} \|\mathbf{m}_t\|^2 + \left( 1 + \frac{2d}{k} \right) \eta_t^2 \mathbb{E} \|\nabla f_{i_t}(\mathbf{x}_t)\|^2 \right) \tag{32}$$

$$\leq \left( 1 - \frac{k}{2d} \right) \mathbb{E} \|\mathbf{m}_t\|^2 + \frac{2d}{k} \eta_t^2 G^2 . \tag{33}$$

By treating the first $t + 1$ iterations specially, we can establish a slightly tighter bound for those. With the inequality $\|\mathbf{a} + \mathbf{b}\|^2 \leq \frac{t}{t-1} \|\mathbf{a}\|^2 + t \|\mathbf{b}\|^2$ for $t \geq 1$ (a consequence of Jensen's inequality), we estimate

$$\mathbb{E} \|\mathbf{m}_{t+1}\|^2 \leq \frac{t+1}{t} \|\mathbf{m}_t - \mathbf{g}_t\|^2 + (t+1) \|\eta_t \nabla f_{i_t}(\mathbf{x}_t)\|^2 \overset{(4)}{\leq} \frac{t+1}{t} \|\mathbf{m}_t\|^2 + (t+1) \eta_t^2 G^2 , \tag{34}$$

and by unrolling

$$\mathbb{E} \|\mathbf{m}_{t+1}\|^2 \leq (t+1) \sum_{i=0}^{t} \eta_i^2 G^2 . \tag{35}$$

Now the claim follows from Lemma A.3 just below with $A = \frac{8G^2}{\mu}$. $\qquad\square$

**Lemma A.3.** *Let $A \geq 0$, $d \geq k \geq 1$, $\{h_t\}_{t \geq 0}$, $h_t \geq 0$ be a sequence satisfying*

$$h_0 = 0 , \qquad h_{t+1} \leq \min \left\{ \left( 1 - \frac{k}{2d} \right) h_t + \frac{2d}{k} \eta_t^2 A , \ (t+1) \sum_{i=0}^{t} \eta_i^2 A \right\} , \tag{36}$$

*for a sequence $\eta_t = \frac{1}{a+t}$ with $a \geq \frac{(\alpha+1)\frac{d}{k} + \rho + 1}{\rho + 1} > 1$, for $\alpha > 4$, $\rho := \frac{4\alpha}{(\alpha-4)(\alpha+1)^2}$. Then*

$$h_t \leq \frac{4\alpha}{\alpha - 4} \eta_t^2 \frac{d^2}{k^2} A , \tag{37}$$

*for $t \geq 0$.*

*Proof.* The claim holds for $t = 0$.

**Large $t$.** Let $t_0 = \max\{\lceil \alpha \frac{d}{k} - a \rceil, 0\}$, i.e. $\eta_{t_0} \leq \frac{k}{\alpha d}$. (Note that for any $a \geq \alpha \frac{k}{d}$ it holds $t_0 = 0$.) Suppose the claim holds for $t \leq t_0$. Observe,

$$\eta_t^2 \left( 1 - \frac{2k}{\alpha d} \right) \leq \eta_{t+1}^2 , \tag{38}$$

for $t \geq t_0$. This follows from Lemma A.2 with $c = \frac{\alpha d}{k}$. By induction,

$$h_{t+1} \leq \left( 1 - \frac{k}{2d} \right) \frac{4\alpha}{\alpha - 4} \eta_t^2 \frac{d^2}{k^2} A + \frac{2d}{k} \eta_t^2 A \tag{39}$$

$$= \underbrace{\eta_t^2 \left( 1 - \frac{2k}{\alpha d} \right)}_{\leq \eta_{t+1}^2} \frac{4\alpha}{\alpha - 4} \frac{d^2}{k^2} A , \tag{40}$$

where we used $t \geq t_0$ (and the observation just above) for the last inequality.

**Small $t$.**  Assume $t_0 \geq 1$, otherwise the claim follows from the part above. We have

$$h_t \leq t \sum_{i=0}^{t-1} \eta_i^2 A \leq \frac{t}{a-1} A \,, \tag{41}$$

where we used

$$\sum_{i=0}^{t-1} \eta_t^2 \leq \sum_{i=0}^{\infty} \frac{1}{(a+i)^2} \leq \int_{a-1}^{\infty} \frac{1}{x^2} \mathrm{d}x = \frac{1}{a-1} \,, \tag{42}$$

for $a > 1$. For $t \leq t_0$ we have

$$\eta_t^2 \frac{d^2}{k^2} \geq \eta_{t_0}^2 \frac{d^2}{k^2} = \frac{1}{(a+t_0)^2} \frac{d^2}{k^2} \geq \frac{1}{\left(\frac{\alpha d}{k}+1\right)^2} \frac{d^2}{k^2} \geq \frac{1}{\left(\frac{(\alpha+1)d}{k}\right)^2} \frac{d^2}{k^2} = \frac{1}{(\alpha+1)^2} \,, \tag{43}$$

using $\frac{d}{k} \geq 1$. Observe $t_0 \leq \alpha \frac{d}{k} - a + 1 \leq (\alpha+1)\frac{d}{k} - a$. For $t \leq (\alpha+1)\frac{d}{k} - a$ we have

$$h_t \leq \frac{t}{a-1} A \leq \frac{(\alpha+1)\frac{d}{k} - a}{a-1} A \leq \rho A \,, \tag{44}$$

by the condition on $a$. Hence, by combining these observations,

$$h_t \leq \frac{t}{a-1} A \leq \rho A = \frac{4\alpha}{\alpha-4} \frac{1}{(\alpha+1)^2} A \leq \frac{4\alpha}{\alpha-4} \eta_{t_0}^2 \frac{d^2}{k^2} A \leq \frac{4\alpha}{\alpha-4} \eta_t^2 \frac{d^2}{k^2} A \,, \tag{45}$$

and the proof follows. $\qquad\square$

*Proof of Lemma 3.3.*  Observe

$$\left(1 - \frac{\mu\eta_t}{2}\right) \frac{w_t}{\eta_t} = \left(\frac{a+t-4}{a+t}\right) \frac{\mu(a+t)^3}{8} = \frac{\mu(a+t-4)(a+t)^2}{8} \leq \frac{\mu(a+t-1)^3}{8} = \frac{w_{t-1}}{\eta_{t-1}} \,, \tag{46}$$

where the inequality is due to

$$(a+t-4)(a+t)^2 = (a+t-1)^3 + \underbrace{1 - 3a - a^2 - 3t - 2at - t^2}_{\leq 0} \leq (a+t-1)^3 \,, \tag{47}$$

for $a \geq 1, t \geq 0$.

We now multiply equation (15) with $\frac{w_t}{\eta_t}$, which yields

$$a_{t+1} \frac{w_t}{\eta_t} \leq \underbrace{\left(1 - \frac{\mu\eta_t}{2}\right) \frac{w_t}{\eta_t}}_{\leq \frac{w_{t-1}}{\eta_{t-1}}} a_t + w_t \eta_t A + w_t \eta_t^2 B - w_t e_t \,. \tag{48}$$

and by recursively substituting $a_t \frac{w_{t-1}}{\eta_{t-1}}$ we get

$$a_T \frac{w_{T-1}}{\eta_{T-1}} \leq \left(1 - \frac{\mu\eta_0}{2}\right) \frac{w_0}{\eta_0} a_0 + \sum_{t=0}^{T-1} w_t \eta_t A + \sum_{t=0}^{T-1} w_t \eta_t^2 B - \sum_{t=0}^{T-1} w_t e_t \,, \tag{49}$$

i.e.

$$\sum_{t=0}^{T-1} w_t e_t \leq \frac{w_0}{\eta_0} a_0 + \sum_{t=0}^{T-1} w_t \eta_t A + \sum_{t=0}^{T-1} w_t \eta_t^2 B \,. \tag{50}$$

We will now derive upper bounds for the terms on the right hand side. We have

$$\frac{w_0}{\eta_0} = \frac{\mu a^3}{8} \,, \tag{51}$$

$$\sum_{t=0}^{T-1} w_t \eta_t = \sum_{t=0}^{T-1} \frac{8(a+t)}{\mu} = \frac{4T^2 + 8aT - 4T}{\mu} \leq \frac{4T(T+2a)}{\mu} \,, \tag{52}$$

and

$$\sum_{t=0}^{T-1} w_t \eta_t^2 = \sum_{t=0}^{T-1} \frac{64}{\mu^2} = \frac{64T}{\mu^2} \,. \tag{53}$$

Let $S_T := \sum_{t=0}^{T-1} w_t = \frac{T}{6}\left(2T^2 + 6aT - 3T + 6a^2 - 6a + 1\right)$. Observe

$$S_T \geq \frac{1}{3} T^3 + \underbrace{aT^2 - \frac{1}{2}T^2 + a^2 T - aT}_{=T^2\left(a-\frac{1}{2}\right) + T(a^2-a) \geq 0} \geq \frac{1}{3} T^3 \,. \tag{54}$$

for $a \geq 1, T \geq 0$. $\qquad\square$

# B Experiments

**Parameter tuning.** To produce a fair comparison between MEM-SGD and QSGD [3], we fix the learning rate to $\gamma_0/(1 + \gamma_0 \lambda t)^{-1}$ and run a grid search on the $\gamma_0$ hyperparameter (individually for each method). The results are displayed in Figure 5.

Figure 5: Hyperparameter search for learning rate $\gamma_0/(1 + \gamma_0 \lambda t)^{-1}$. $\gamma_0$ corresponding to each lowest curve are used in section 4.3. *Top row:* RCV1-test dataset. *Bottom row:* epsilon dataset

**QSGD communicated bits.** The number of bits needed by QSGD with $s$ quantization levels to communicate the gradient at each iteration is $\min\{(\lceil \log_2(s) \rceil + 1) \cdot d, 3s(s + \sqrt{d}) + 32\}$ where $d$ is the size of the gradient. The first expression corresponds to the naïve encoding (i.e. index/value pairs), the second expression corresponds to the estimates of the more evolved Elias encoding (see e.g. [3, Theorem 3.2]). For the sparse dataset *RCV1-test*, we additionally assume that QSGD is aware of the sparsity of the gradients ($d \approx 71 \ll 47'236$) and send only the quantized non zero coordinates with their indexes. In a nutshell, we chose the best communication pattern for QSGD to conduct a fair comparison with MEM-SGD.