[Reviews · NeurIPS 2018]

Reviewer 1



This is a fairly complete paper establishing the utility of a way to reduce communication overhead by a simple technique to sparsifying gradients and update a shared memory [in parallel]. It proves a convergence bound and establishes that parameters of the algorithm do in fact have reasonable values, leading to a practical algorithm. Benefits are demonstrated experimentally. The proof is understandable. Note that just as for another submission, Lemma 3.3 has an 'e_t'. However this time the proof makes sense, because e_t appears in both eq. 15 and 16 (This is a major improvement on the other paper, which I guess has some typo and led to complete confusion when reading through the proof.) Reading this paper was a pleasant experience, and the technique is fairly simple to implement. Providing experimental demonstrations [even if non-convex] also allows people less interested in theory to appreciate this paper.

Reviewer 2



The paper talks about the convergence analysis of a specific type of SGD for convex problems, where the gradients are k-sparsified: either top-k or random chosen. It provides proofs that the proposed algorithm converges at the same rate as vanilla SGD. The authors claim that it is the first convergence result for sparsified SGD with memory. The numerical study verifies the convergence theory, shows its effectiveness comparing with QSGD in terms of performance and memory requirements. Also, MEM-SGD scales up to 10 cores. The results are overall encouraging. In Page 6, line 179, in the sentence “As the usefulness of the scheme has already been shown in practical applications“, what is the scheme? Is it MEM-SGD? Or Quantization? Reducing communication overhead is mostly interesting in training deep learning models. Can this work be extended to non-convex setting? As to the memory containing pass information, it is better to do an experiment to comparing the results of using or not using the memory.

Reviewer 3



This work analyzes a variant of SGD designed to incur lower communication costs in distributed systems. This variant of SGD combines sparsification with memory. At each iteration, the algorithm sparsifies the stochastic gradient by sending only the k (out of d) largest coordinates (in absolute value) or a random k coordinates to other nodes in the system. The portion that was not transmitted is added to the next stochastic gradient, and their sum is sparsified in the same manner. The authors show that without memory, the convergence of this method could be adversely affected, but that by maintaining this memory, the convergence rate resembles that of SGD, modulo a d/k factor. The main theory of the paper uses perturbed iterate analysis to analyze sparsified SGD with memory. The authors also give detailed experiments showing that selecting the top k coordinates outperforms random k coordinates, and comparing the method favorably to QSGD. Quality: The submission seems to be theoretically sound. Many of the techniques are straightforward adapatations of techniques from perturbed iterate analysis. The primary theoretical novelty is Lemma 3.2 which uses the fact that sparsification can be seen as a kind of contraction operator (Definition 2.1) and analytic estimates to analyze the norm of the memory vector over time. Moreover, the experimental results show (as prior works have shown) the efficacy of the strategy. I have only two main comments about this part. The first is that some discussion of how this analysis might carry over to the convex or even non-convex setting would be illuminating. Seeing where and how the theory breaks could help spur the community to fix these issues. The second has to do with Figure 3. The authors correctly note that sparsified SGD with memory can be much more sparse than QSGD. The authors try 2, 4, and 8 bit QSGD (as the authors of the original QSGD paper do), but they only display 4 and 8 bit QSGD for one experiment and 2 and 4 for the other. It is not immediately clear why this omission occurs. Clarity: The paper is well-written throughout. The underlying problem is well-motivated in the introduction, and a very useful technical motivation is given in Section 2.2. Comparing the variance blow-up reduction to that of mini-batch SGD was particularly insightful. The theory is organized well, and the experiments setups are detailed. A few comments that might help the readability throughout. First, more clearly differentiating between sparsification and quantization may be helpful. For instance, in line 29 you refer to a quantization function as sparse(), even though quantizations need not be sparse. Second, in the appendix every equation is numbered. It might be more helpful to the reader to only number the important ones. Intermediate steps need not be numbered. Originality: The theoretical results in the work are to the best of my knowledge novel. They are very similar in scope to results concerning the use of 1-bit SGD with memory, but apply to sparsification instead. The authors do a good job of citing other work that has analyzed SGD with memory empirically and differentiate themselves by presenting their theoretical work. While the authors do a good job of citing all the related work, some of the related work section could be fleshed out or improved. For instance, lines 51-52 state that "Sparsification methods reduce the number of non-zero entries in the stochastic gradient," which while true, ignores all nuance in different methods. Short general explanations of what some of the methods do would be useful, especially for QSGD which is later compared directly to SGD with memory. Last, some of the phrasing in the related work section is based on opinion or conjecture instead of fact. For instance, line 55 says that sparsification methods "can only work" when memory is incorporated. While I agree that vanilla sparsification methods can use memory to their benefit, to suggest they are never effective without them or that there are no other possible methods is not necessarily correct. Significance: The theory is useful for confirming what empirical studies have seen in practice. The theory is also significant for its simplicity and conciseness. In terms of scope, it is similar to other results (such as 1-bit SGD with memory), and only extends to strongly convex functions. Nevertheless, its mathematical elegance is worth highlighting. Its experimental results comparing SGD with memory to QSGD are somewhat significant as well, though due to the number of variants of QSGD and parameters to be set, a more comprehensive analysis might be warranted. All in all, I believe that the paper deserves acceptance to NIPS. It is well-written, mathematically concise, and gives a step forward in understanding how the use of memory can help sparsification/quantiztion techniques.